# Silent Hypoxemia in the Emergency Department: A Retrospective Cohort of Two Clinical Phenotypes in Critical COVID-19

**DOI:** 10.3390/jcm11175034

**Published:** 2022-08-27

**Authors:** Karine Alamé, Elena Laura Lemaitre, Laure Abensur Vuillaume, Marc Noizet, Yannick Gottwalles, Tahar Chouihed, Charles-Eric Lavoignet, Lise Bérard, Lise Molter, Stéphane Gennai, Sarah Ugé, François Lefebvre, Pascal Bilbault, Pierrick Le Borgne

**Affiliations:** 1Emergency Department, Hôpitaux Universitaires de Strasbourg, 67000 Strasbourg, France; 2CREMS Network (Clinical Research in Emergency Medicine and Sepsis), 67201 Wolfisheim, France; 3Emergency Department, Regional Hospital of Metz-Thionville, 57700 Hayange, France; 4Emergency Department, Mulhouse Hospital, 68100 Mulhouse, France; 5Emergency Department, Colmar Hospital, 68000 Colmar, France; 6Emergency Department, Nancy University Hospital, 54000 Nancy, France; 7Centre d’Investigations Cliniques-1433, and INSERM U1116, F-CRIN INI-CRCT, Université de Lorraine, 54000 Nancy, France; 8Emergency Department, Nord Franche Comté Hospital, 90400 Trévenans, France; 9Emergency Department, Haguenau Hospital, 67504 Haguenau, France; 10Emergency Department, Verdun Hospital, 55100 Verdun, France; 11Emergency Department, Reims University Hospital, 51100 Reims, France; 12Department of Public Health, University Hospital of Strasbourg, 75016 Paris, France; 13INSERM (French National Institute of Health and Medical Research), UMR 1260, Regenerative NanoMedicine (RNM), Fédération de Médecine Translationnelle (FMTS), University of Strasbourg, 75016 Paris, France

**Keywords:** COVID-19, phenotypes, silent hypoxemia, happy hypoxemia, critical care, acute respiratory distress syndrome

## Abstract

Introduction: Understanding hypoxemia, with and without the clinical signs of acute respiratory failure (ARF) in COVID-19, is key for management. Hence, from a population of critical patients admitted to the emergency department (ED), we aimed to study silent hypoxemia (Phenotype I) in comparison to symptomatic hypoxemia with clinical signs of ARF (Phenotype II). Methods: This multicenter study was conducted between 1 March and 30 April 2020. Adult patients who were presented to the EDs of nine Great-Eastern French hospitals for confirmed severe or critical COVID-19, who were then directly admitted to the intensive care unit (ICU), were retrospectively included. Results: A total of 423 critical COVID-19 patients were included, out of whom 56.1% presented symptomatic hypoxemia with clinical signs of ARF, whereas 43.9% presented silent hypoxemia. Patients with clinical phenotype II were primarily intubated, initially, in the ED (46%, *p* < 0.001), whereas those with silent hypoxemia (56.5%, *p* < 0.001) were primarily intubated in the ICU. Initial univariate analysis revealed higher ICU mortality (29.2% versus 18.8%, *p* < 0.014) and in-hospital mortality (32.5% versus 18.8%, *p* < 0.002) in phenotype II. However, multivariate analysis showed no significant differences between the two phenotypes regarding mortality and hospital or ICU length of stay. Conclusions: Silent hypoxemia is explained by various mechanisms, most physiological and unspecific to COVID-19. Survival was found to be comparable in both phenotypes, with decreased survival in favor of Phenotype II. However, the spectrum of silent to symptomatic hypoxemia appears to include a continuum of disease progression, which can brutally evolve into fatal ARF.

## 1. Introduction

The ongoing pandemic of Coronavirus disease-19 (COVID-19) has been overwhelming the world for the past two years. After it first emerged in Wuhan, China, in December 2019, the novel coronavirus at cause was identified as severe acute respiratory syndrome-coronavirus-2 (SARS-CoV-2) [1]. As of July 2022, the outbreak has cumulated in over 550 million confirmed COVID-19 cases, with over 6.3 million deaths worldwide [2].

SARS-CoV-2 seems to infect the host’s airways by predominantly binding with the angiotensin-converting enzyme-2 (ACE2) for cell entry, a receptor broadly distributed on various tissue and immune cells, correlating with the range of COVID-19 symptoms and multiorgan dysfunction [3]. Subsequently, the progression of infection results from the interplay of the direct cytopathic effects of the virus [3], the activation of immune-mediated pathways [4], a dysregulation of the immune system leading to a cytokine release syndrome [5], and coagulopathy and vascular dysfunction, including an interplay between immunity and coagulation termed immunothrombosis [6,7]. Therefore, SARS-CoV-2 can generate diverse clinical manifestations. Most patients are asymptomatic or pauci-symptomatic, presenting influenza-like signs, several of which develop mild disease and require hospitalization for viral hypoxemic pneumonia. A minority of patients present critical disease with complications such as COVID-19 related ARDS (CARDS) [8,9].

Since the early days of the COVID-19 pandemic, a peculiar respiratory phenomenon was observed, both perplexing and bewildering first-line physicians [10,11]. Some critical COVID-19 patients presented acute respiratory failure (ARF), associating this with severe hypoxemia with clinical signs of respiratory distress that satisfied standard ARDS criteria. Whereas other patients presented profound arterial hypoxemia yet showed a lack of any clinical sign of respiratory distress and appeared cooperative and seemingly comfortable, conversing and scrolling on their phones [9]. This state was termed ‘silent’, ‘happy’, or ‘non-dyspneic’ hypoxemia [12]. For first-line and bedside clinicians, the atypical presentation created a serious distraction regarding important decisions, such as the timing of endotracheal intubation, ventilation strategies, and orientation to the ward or the ICU [13,14].

Two years after SARS-CoV-2 emerged, the literature exhibits varying theories, some speculating and others validating the mechanisms behind the dissociation between profound hypoxemia and the lack, or presence, of clinical signs of ARF in some patients. To some authors, CARDS is an atypical form of ARDS, suggesting its clinical presentation to be the result of ventilation-perfusion impairments [8,11]. Others associate it with the neuro-invasive potential of the virus [15]. Further hypotheses include chemoreflex dysregulation similar to high-altitude exposure and other hypoxic response and decline mechanisms [16,17].

Yet, beyond speculation on the pathophysiology behind these two distinct clinical phenotypes, only a few studies document the baseline parameters and outcomes for severe to critical COVID-19 [18]. Understanding this presentation in critically ill patients is key for their proper and timely management. Therefore, in a population of critical COVID-19 patients admitted to the emergency department (ED), we aimed to study the clinical characteristics, management, and outcomes of silent hypoxemia (Phenotype I) in comparison with symptomatic hypoxemia with clinical signs of ARF (Phenotype II) along with the factors associated with each presentation.

## 2. Methods

### 2.1. Study Settings

This is a retrospective cohort study, conducted in the Great-East region of France, an area heavily impacted by the COVID-19 pandemic in Europe during the first wave. This multicenter study was led with the participation of nine French hospitals: three university hospitals (CHRU of Strasbourg, CHRU of Nancy, and CHU of Reims) and six general hospitals (Colmar Hospital, Haguenau Hospital, Mulhouse Hospital, Metz-Thionville Hospital, Nord Franche-Comté Hospital, and Verdun Hospital).

### 2.2. Study Population

Between 1 March and 30 April 2020, during the first wave of the COVID-19 outbreak, all adult patients who were presented to the emergency department (ED) of these nine Great-Eastern French hospitals for confirmed, severe, or critical COVID-19, and were then directly admitted to the intensive care unit (ICU), were included in this study. In concordance with current guidelines and WHO definitions, severe COVID-19 was defined by patients with a respiratory rate of 30 cycle/min or more and a hemoglobin oxygen saturation of 90 to 93% or less. Critical COVID-19 was defined as the occurrence of complications such as ARDS and thromboembolism [19]. Diagnosis of SARS-CoV-2 infection was laboratory confirmed by RT-PCR using nasopharyngeal specimens. Patients with no laboratory confirmed COVID-19 diagnosis, along with those suffering from mild to moderate disease and those who received ambulatory care or in-hospital care in a conventional medical unit, together with those who were secondarily admitted to the ICU, were all excluded from the study. Patients who were subject, during ED management, to limitation of therapeutic effort (including efforts of withdrawing or withholding life-sustaining therapy) were also excluded.

In early 2020, authors first postulated the presence of two primary clinical phenotypes in critical COVID-19 [10,11]. Phenotype I corresponds to patients with silent hypoxemia, presenting no signs of acute respiratory failure (ARF), and Phenotype II corresponds to symptomatic patients with both signs of hypoxemia (cyanosis, impaired consciousness, blood oxygen saturation below 90%) and ARF (respiratory rate above 30 cycle/min or below 15 cycle/min, signs of hypercapnia, diaphoresis, use of accessory respiratory muscles such as sternocleidomastoid contraction, intercostal retraction or paradoxical motion of the abdomen).

### 2.3. Data Collection

Electronic medical records were retrospectively studied then queried for demographical, clinical, and biochemical data, which were standardized in a report file. We recorded primary epidemiological factors such as age and sex, along with essential comorbidities such as obesity, (body weight mass over 30 kg/m^2^), history of cardiovascular or respiratory disease, diabetes, pre-existing renal failure, and any history of malignancy or immunodeficiency. Different aspects of ED management were documented, such as clinical parameters and the call for early organ support strategies, including endotracheal intubation. Laboratory results such as arterial blood gas, creatinine, and C-reactive protein, were also collected. Radiological findings were documented, mainly reporting on the extension of lesions. The severity of the illness was determined using the simplified acute physiology score (SAPS II) [20]. Arterial oxygen partial pressure to fractional inspired oxygen (PaO_2_/FiO_2_) was measured within the first 24 hours of ICU admission, and ARDS was classified according to the Berlin definition [21]. Ventilation strategies in the ICU were documented, reporting on the call for emergency endotracheal intubation, the use of the prone position, and the duration of mechanical ventilation. Different ICU organ support strategies were also documented, including the use of extracorporeal membrane oxygenation, renal replacement therapy, vasopressor drug support, and continuous muscle blockers. Additionally, the occurrence of secondary complications such as thromboembolic events was reported. At last, after comparing the two clinical phenotypes, we studied mortality and length of stay in the ICU and in hospital. Survival follow-up was obtained for the entire study population, which allowed us to generate a survival curve, visualizing mortality over 120 days.

### 2.4. Statistical Analysis

Our analysis included both a descriptive and an analytical section. We performed a descriptive statistical analysis of the quantitative variables by giving frequencies and percentages for categorical variables. Analysis of the continuous variables was performed by giving the median along with the first and third quartiles and means, along with its standard deviations. Two-group comparisons of continuous covariates were performed by Mann–Whitney-U test. Comparisons between the categorical variables were made using Chi-squared test or Fisher’s exact test in case of expected values in any of the cells of a contingency table were below 5. A multivariate logistic model was then performed on the statistically significant and clinically relevant covariates. For survival analysis, time-to-event curves were computed with the use of the Kaplan–Meier method and were compared using log-rank tests. Analyses were performed with R 4.0.2 software (R Foundation for Statistical Computing, Vienna, Austria) in its most up-to-date version, as well as with all the software packages required to carry out this statistical investigation. The *p* values were two-sided, and statistical significance was set at *p* < 0.05.

## 3. Results

### 3.1. Study Population

Between 1 March and 30 April 2020, the ED of all nine participating hospitals witnessed a total of 72,941 patient visits, out of whom 9296 (12.7%) were diagnosed with COVID-19. The ICU received and managed 776 of these patients. Conclusively, after excluding patients with missing data and those secondarily admitted to the ICU, a total of 423 patients were included in this study (Figure 1).

### 3.2. Demographics, Clinical Characteristics and Management

The median age of our cohort was 66 years (58–72 years), and most patients were male (73.5%, CI 95%: 69–77.7%). When admitted to the ED, most patients presented with critically altered clinical respiratory parameters. Subsequently, the median respiratory rate was 30 cycles/min (24–35 cycles/min), and the median blood oxygen saturation was 90% (84–94%). Over half of the study population showed typical clinical signs of respiratory failure (56% Phenotype II, CI 95%: 51.1–60.8%), while the remaining 44% (Phenotype I, CI 95%: 39.2–48.9%) showed none. In regards to the laboratory findings, arterial blood gas revealed a median oxygen tension of 67 mmHg (55–81 mmHg) and a median carbon dioxide tension of 34 mmHg (30–38 mmHg). Most patients (85.3%, CI 95%: 75.8–95%) were intubated within the first 20 hours of management, of which around 53.3% occurred in the ICU compared to 46.7% in the ED. According to the Berlin definition, most of the study population (91%, CI 95%: 87.7–93.5%) satisfied ARDS criteria within the first 24 h of their ICU stay. Overall, mechanical ventilation lasted for a median of 14 days (7–24 days). In regards to organ support strategies in the ICU, most patients were treated with prone ventilation (62.5%, CI 95%: 57.6–67.1%), continuous muscle blockers, and vasopressor drugs (77.4% and 78.2%, respectively). Secondary complications, such as thromboembolic events, were reported in 16.2% (CI 95%: 12.8–20%) of cases. Lastly, hospital stay lasted for a median of 26 days (13–43 days) in total, including 17 days (8–30 days) in the ICU. Overall patient characteristics are summarized in Table 1.

### 3.3. Survival Status

A total of 423 critical to severe COVID-19 patients were included in our study, out of whom a quarter (26.5%, CI 95%: 22.3–31.0%) did not survive the hospital stay. When comparing survivors and non-survivors, survivors were significantly younger (64 years vs. 69 years, *p* < 0.001). Non-survivors presented more clinical signs of respiratory distress (*p* = 0.002). Initially, during ED management, the latter presented profound hypoxia with lower blood oxygen saturation levels (88% vs. 91% for survivors, *p* = 0.005) and hypoxemia (63 mmHg vs. 68 mmHg, *p* = 0.037), corresponding to Phenotype II patients, as opposed to survivors who corresponded further to a silent hypoxemia phenotype. Other clinical and biochemical, along with radiological findings, did not differ according to survival status. Non-survivors were presented more significantly in a state of ARDS with a PaO_2_/FiO_2_ < 300 mmHg (96.3% vs. 89%, *p* = 0.024). Subsequently, in the non-survivor subgroup, vasopressor support was increased (*p* = 0.013), and endotracheal intubation was performed mostly in the ED (*p* = 0.002). In the ICU, prone ventilation was further used on non-survivors compared to survivors (*p* = 0.022). Consequently, survivors were subjected to a more prolonged duration of both mechanical ventilation (*p* = 0.0014), ICU stay (*p* < 0.001), and hospital stay (*p* < 0.001). Overall survival findings are summarized in Table 1.

### 3.4. Phenotype I versus Phenotype II: Characteristics and Management

In total, over half of the patients presented a state of profound and symptomatic hypoxemia with clinical signs of ARF (56.1%), whereas the rest presented a state of silent hypoxemia (43.9%). Regarding demographical findings, no significant difference was found in the two subgroups. However, the population demonstrated some significant differences regarding comorbidities; patients presenting silent hypoxemia also presented significantly less history of cardiovascular diseases (*p* = 0.029), malignancies, and immunodeficiency (*p* = 0.041). Clinically, compared to silent hypoxemia, patients displaying symptomatic hypoxemia with clinical signs of ARF presented significantly lower blood oxygen saturation (89% vs. 91%, *p* < 0.001), a higher respiratory rate (30/min vs. 26/min, *p*< 0.001), and a higher oxygen requirement (15 L/min vs. 12 L/min, *p* < 0.001). Arterial blood gas analysis revealed a comparable hypoxemia/hypocapnia syndrome in both phenotypes I and II. The partial pressure of oxygen and carbon dioxide leveled at 67 mmHg and 34 mmHg, respectively, for both clinical phenotypes, yet pH was significantly lower in phenotype II patients (*p*< 0.001). Those patients also presented more extended pulmonary lesions in chest CT scans (95% vs. 45%, *p* < 0.001). Subsequently, patients with Phenotype II were managed with early lung protective ventilation strategies, where patients were primarily intubated initially in the ED (46% vs. 26.9%, *p* < 0.001). Whereas patients with silent hypoxemia were, for the majority, intubated secondarily in the ICU (56.5% vs. 35.9%, *p* < 0.001). No significant difference was found regarding the severity of ARDS between the two subgroups within the first 24 h of management. The use of organ support strategies in the ICU did not significantly differ between both phenotypes, as was the case for the duration of mechanical ventilation and ICU stay. The overall phenotype findings are summarized in Table 2.

### 3.5. Phenotype I versus Phenotype II: Survival Status

The initial univariate analysis revealed higher ICU mortality (29.2% vs. 18.8%, *p* < 0.014) and in-hospital mortality (32.5% vs. 18.8%, *p* < 0.002) in patients presenting clinical respiratory distress along with profound hypoxemia and ARF, compared to those presenting silent hypoxemia. A survival curve was generated for these two subgroups, allowing for the visualization and comparison of their survival over 120 days (Figure 2, log-rank *p* = 0.004). Survival at 60 days was lower for patients in clinical respiratory distress hypoxemia compared to those with Phenotype I, with a survival rate of around 59% and 75%, respectively. At 120 days, the survival rate similarly leveled at 75% for silent hypoxemia patients, yet it decreased to around 40% for Phenotype II. However, when comparing the mortality rates of both phenotypes, multivariate analysis revealed no significant difference both in-ICU mortality (aOR = 2.385, CI 95%: 0.715–7.950, *p* = 0.157) and in-hospital mortality (aOR = 3.079, CI 95%: 0.932–10.171, *p* = 0.065). Excessive mortality might be further associated with Phenotype II patients. The survival findings are summarized in Figure 2 and Table 3.

### 3.6. Phenotype I versus Phenotype II: Multivariate Analysis

When comparing the adjusted values, a multivariate analysis of both of the clinical phenotypes demonstrated that nearly all demographical characteristics and comorbidities were similar in the two subgroups, with no significant difference. In regards to ED management, the two phenotypes presented comparable clinical parameters, except for respiratory rate (aOR = 1.102, CI 95%: 1.043–1.166, *p* = 0.001), which was evidently significantly associated with Phenotype II. In regards to imaging, our results displayed pulmonary lesion extensions of over 50% for chest-CT scans (aOR = 3.017, CI 95%: 1.270–7.166, *p* = 0.012) as being more associated with Phenotype II patients. Yet, the typical COVID-19 aspect of the lesions (aOR = 0.331, CI 95%: 0.128–0.857, *p* = 0.023) was more associated with silent hypoxemia patients. In regards to patient management in the ED, endotracheal intubation (aOR = 3.844, CI 95%: 1.199–12.319, *p* = 0.023) was further performed when patients presented signs of ARF. Adversely, when comparing the duration of mechanical ventilation, different ICU organ support strategies, mortality, and length of stay (in the ICU and in-hospital), no significant differences were found between the two subgroups. The overall multivariate analysis findings are summarized in Table 3.

## 4. Discussion

### 4.1. Two Clinical Phenotypes in Critical COVID-19

During the COVID-19 pandemic, a prevalent feature of the disease astounded physicians and gained extensive coverage. Our findings appear straight forward and consistent with the literature. In the ED, some patients were reported to have an atypical clinical presentation, progressing towards a form ARDS, termed CARDS, which was characterized by severe hypoxemia; however, they showed near-normal respiratory mechanics and high lung gas volume, high lung compliance, and minimal alveolar recruitability, despite having low PaO_2_/FiO_2_ [11,22]. Other patients were reported with a clinical presentation progressing towards a more typical form of ARDS, characterized by severe hypoxemia and critically altered respiratory mechanics, with low lung gas volumes with intrapulmonary shunt, high lung weight, high alveolar recruitability, and minimal compliance [11,23]. *In fine*, the 423 critical COVID-19 patients studied in our cohort were all hypoxemic and were stratified according to the aforementioned clinical phenotypes: over half of them presented in Phenotype II, whereas the rest presented in Phenotype I, despite profound hypoxemia and a low PaO_2_/FiO_2_ ratio.

At that time, the initially presumed novel presentation created serious distraction regarding major decisions such as patient triage and timely treatment, especially the timing of intubation, which is an added challenge to healthcare systems facing times of crisis. In reflection of the peak phase of the COVID-19 pandemic, our work exhibits the role of initial clinical presentation and its assessment in disease progression. This is of interest to better understand the trajectory of this disease and its outcomes (intubation and mortality) in critically ill patients. This understanding might help clinicians avoid any delay in the assessment of COVID-19 progression, make timely treatment decisions, especially on a ventilatory level, and accurately triage, determining the correct patient orientation and level of care.

This heterogeneity of COVID-19 presentation during the first waves of the pandemic, ranging from silent hypoxemia to severe signs of acute respiratory failure, might be time-related to the evolution of pathophysiologic changes in different stages of the disease [18]. In our multivariate analysis, an independent predictor of hypoxemia with clinical signs or respiratory failure was the respiratory rate. Elevated respiratory rates and increased lung mechanics in Phenotype II patients compared to those with silent hypoxemia, despite the same PaCO_2_ (median 34 mmHg), is a potential sign of increased pulmonary dead space, corresponding to a transition to a severe outcome with a tendency for decreased survival in Phenotype II. Therefore, silent to symptomatic hypoxemia in COVID-19 patients is more associated with a continuum of the disease rather than with the two distinct phenotypes, where an increase in respiratory mechanics and the appearance of clinical signs of respiratory distress might indicate a turning point in disease evolution and require the rapid escalation of ventilation support.

### 4.2. In the Times of Pandemic Crisis

The COVID-19 pandemic drowned medical systems worldwide under endless streams of pressure and challenges. Emergency and critical care systems were overwhelmed by surges of critically ill, deteriorating patients in need of adequate triage and intensive care. However, the resources were inadequate, forcing hospitals to undergo a structural reorganization to accommodate a time of crisis [24,25]. In uncharted medical territory, the shortage of reserves during the peak stages of the pandemic created vital dilemmas about ideal rationing and timely treatment. Healthcare systems were forced to develop mass triage strategies and crisis standards for care plans and the allocation of resources [26]. In the times of the pandemic crisis, decisions regarding patient triage were further clouded by this peculiar feature of COVID-19, prevalent in our cohort: silent and symptomatic hypoxemia.

At the time, patients were mainly managed following the current guidelines; there was no alternative to prolonged mechanical ventilation and patient treatment relied on typical ARDS therapeutic strategies since the potential clinical benefits of combined therapy using non-invasive ventilation were not yet validated due to the potential aerosolization of the then emerging, and unknown, viral particles [27]. As elaborated in our analysis, when comparing different ED and ICU organ support strategies, no significant differences were found between the two subgroups other than the timing of endotracheal intubation, which was further performed earlier in the ED for Phenotype II. At that time, patient triage, timely treatment, the timing of intubation, and ICU admission remained a matter of debate in patients depending on their severity [13,28]. Yet, delaying their timely treatment was associated with a poorer prognosis and increased mortality, partly because delaying the proper and timely treatment of inflamed lungs in hypoxemic patients with vigorous spontaneous respiratory effort promotes self-inflicted lung injury [29]. In parallel, high demand on critical and emergency services also increased mortality for patients with critical COVID-19, both in the early and later stages of the pandemic [30].

### 4.3. Beyond Silent Hypoxemia

Authors found that the dissociation between the profound level of hypoxemia and the lack of clinical signs of respiratory distress was first associated with an imbalance between the processes inducing hypoxemia at the beginning of the disease and the initially preserved lung mechanics with no increased airway resistance or dead space ventilation, hence not stimulating the respiratory centers [31]. Yet, the mechanisms underlying oxygenation impairment in COVID-19 patients seem to primarily be the result of a mismatch between lung ventilation and perfusion ratio, which depends on the adequacy of gas exchange [8,11]. The gas exchange impairment of severe COVID-19 patients is attributable to substantial endothelial damage, shunt due to gasless tissue as in all-cause ARDS, and a pathological lung hyperperfusion caused by thrombosis of diseased poorly ventilated lung region [6,7,32]. Consequently, this mechanism leads to silent hypoxemia, progressing later towards symptomatic manifestation [32].

Another mechanism behind this clinical presentation is attributable to the idiosyncratic action of SARS-CoV-2 on the receptors involved in chemosensitivity to oxygen [8,32]. Respiratory response to a decrease in oxygen blood tension is quantified by the hypoxic ventilatory response, a fundamental physiological response to hypoxia, which aligns with physiology, regardless of the cause of hypoxemia, and is largely mediated by the carotid chemoreceptors and regulated based on pCO_2_ [11,12,13,14,15,16,17]. Hypocapnia induces respiratory alkalosis, increasing arterial oxygen saturation [16,17]. Hypoxemia also stimulates carotid-body chemoreceptors, increasing the respiratory drive, in contrast to the depressant effect of acute hypercapnia on the central respiratory drive [16,17]. This is modeled by Ottestad et al. [16] in aviation medicine and hypobaric chamber experiments, revealing that hypocapnic hypoxia is not usually accompanied by air hunger, which is all due to hypoxemia/hypocapnia syndrome resulting in silent hypoxemia [16]. In parallel, the lack of respiratory distress is partly accounted for by hypoxic ventilatory decline, which is meditated by rapidly changing inhibitory responses to hypoxemia [17]. Other mechanisms, such as the viral invasion of the central nervous system, have been put forward to explain silent hypoxemia, although the scientific evidence is lacking [15]. Nevertheless, as discussed, according to the physiological hypoxic ventilatory response, the absence of respiratory distress despite severe hypoxemia is not specifically linked to COVID-19 but to other lung diseases as well [17]. However, the silent hypoxemia phenomenon, which is recorded in nearly 10% of patients with non-COVID-19 related ARDS, seems to be more prevalent in CARDS [33].

### 4.4. Limitations

This study has several limitations. First, it’s retrospective nature since it was difficult to build a prospective cohort in the peak phase of the outbreak. Second, the collected data could not be exhaustively detailed, owing to crisis circumstances. Third, the size of our population is relatively small. Consequently, our results might lack statistical power. We are aware of the limitations of a study based on the role of silent hypoxemia in critical patients, in which a lot of exhaustive data about ventilation parameters may be lacking. Now, two years after its onset, and seemingly due to all the novel virus variants, silent hypoxemia appears to be less prevalent today than it was when the virus first emerged, and different stages of the disease are now faced differently, especially in their severe to critical forms. High-flow nasal oxygen and non-invasive ventilation, along with awake-prone positioning and careful monitoring, are widely used nowadays to delay and avoid prolonged mechanical ventilation in COVID-19 patients. Regardless of the limitations, our first-line clinical observations are clear; our research brings awareness to these phenotypes, along with knowledge of the events that unfolded during the first stages of the COVID-19 pandemic. This may assist clinicians and shed light on making ICU admission decisions and hence help to better face the next health crisis when the system will be undoubtedly overflowing with surges of patients again.

## 5. Conclusions

Two clinical phenotypes were prevalent for critical COVID-19 in our cohort ever since the first wave of the pandemic: silent hypoxemia and symptomatic hypoxemia. The dissociation between a profound level of hypoxemia and the lack, or presence, of clinical signs of respiratory distress, which can mislead clinicians, can be explained by various mechanisms, most of them physiological and unspecific to COVID-19. We found survival to be comparable in both phenotypes, with decreased survival in favor of patients with symptomatic hypoxemia. However, the spectrum of silent to symptomatic hypoxemia, rather than being two distinct clinical presentations, appears to be included in a continuum of disease progression, which can brutally evolve into fatal acute respiratory failure. Hence, patients with silent hypoxemia should be closely monitored as they can rapidly become critical.

## Figures and Tables

**Figure 1 jcm-11-05034-f001:**
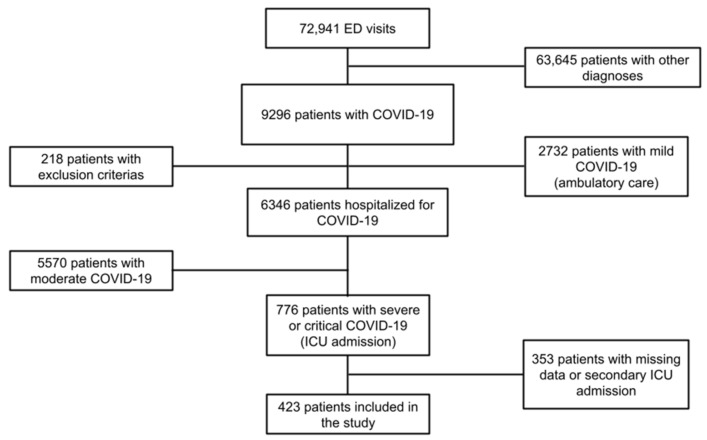
Flowchart of the study. Abbreviations: ED = emergency department, ICU = intensive care unit, COVID-19 = Coronavirus disease 2019.

**Figure 2 jcm-11-05034-f002:**
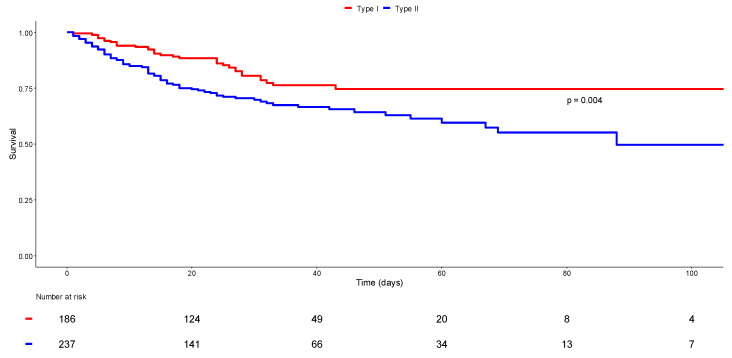
Survival status according to clinical phenotype. Kaplan–Meier curves display the survival probability of patients with silent hypoxemia (Phenotype I in red) and those with hypoxemia with clinical signs of acute respiratory failure (Phenotype II in blue). Log-range *p* = 0.04.

**Table 1 jcm-11-05034-t001:** Demographics, clinical characteristics, and management, according to survival status.

Demographics	All Patients *n* = 423	Survivors *n* = 311	Non-Survivors *n* = 112	*p*-Value
Age (years)	66 [58–72]	64.0 [56–71]	69 [64–74.3]	**<0.001 ***
Male	311 (73.5)	222 (71.4)	89 (79.5)	0.096
**Comorbidities**				
Hypertension	235 (55.6)	164 (52.7)	71 (63.4)	0.052
BMI > 30 (kg/m^2^)	174 (42.1)	131 (43.0)	43 (39.8)	0.571
Cardiovascular diseases	135 (31.9)	89 (28.6)	46 (41.1)	**0.015 ***
Diabetes mellitus	118 (27.9)	82 (26.4)	36 (32.1)	0.243
Pre-existing renal failure	69 (16.5)	45 (14.6)	24 (22.0)	0.071
Malignancies or ID	52 (12.3)	35 (11.3)	17 (15.2)	0.278
Respiratory diseases	95 (22.5)	69 (22.2)	26 (23.2)	0.823
**ED management**				
Respiratory rate (/min)	30 [24–35]	29 [24–35]	30 [24.5–35.5]	0.218
First O_2_ saturation (%)	90 [84–94]	91 [85–95]	88 [82–93]	**0.005 ***
O_2_ requirement (L/min)	15 [6–15]	15 [6–15]	15 [9–15]	0.085
Heart rate (/min)	93 [82–107]	93 [83–106.3]	94 [79–107]	0.845
Systolic BP (mmHg)	130 [115–142]	130 [115.8–142]	129 [113–145]	0.364
Glasgow score scale	15 [15–15]	15 [15–15]	15 [15–15]	0.213
Temperature (°C)	37.8 [37–38.6]	37.8 [37–38.6]	37.9 [36.9–38.5]	0.719
Duration since onset sympt (days)	7 [5–10]	7 [5–10]	7.0 [4–8]	**0.007 ***
Intubation in the ED	159 (37.6)	103 (33.1)	56 (50.0)	**0.002 ***
Phenotype I	186 (44.0)	151 (48.6)	35 (31.3)	**0.002 ***
Phenotype II	237 (56.0)	160 (51.4)	77 (68.7)	**0.002 ***
**Laboratory findings**				
Creatinine (μmol/L)	84 [67–105]	80 [66–101.9]	93.0 [72–119]	**0.001 ***
Lymphocytes (/μL)	780 [580–1110]	790 [600–1127.5]	725 [500–1063]	0.144
CRP (mg/L)	148.2 [83–223]	147.2 [85.4–222.3]	153 [79–223]	0.766
pH	7.46 [7.42–7.49]	7.46 [7.42–7.49]	7.46 [7.41–7.49]	0.456
PaO_2_ (mmHg)	67 [55–81]	68 [58–82]	63 [53–78]	**0.037 ***
PaCO_2_ (mmHg)	34 [30–38]	34 [30–38]	33 [28–37]	0.116
HCO_3-_ (mmol/L)	23.5 [21.4–25.9]	23.8 [22.0–26.0]	23.0 [19.8–25.0]	0.009
Lactate (mmol/L)	1.4 [1.1–2]	1.4 [1–1.9]	1.6 [1.2–2.4]	**0.002 ***
**Radiological findings**				
Typical CT-scan	223 (53.5)	170 (55.6)	53 (47.8)	0.158
Extension > 50%	140 (45.9)	107 (46.1)	33 (45.2)	0.891
**ICU management**				
SAPS II	42 [32–54]	40 [30.5–51]	47 [39–58]	**<0.001 ***
ARDS	372 (91.0)	268 (89.0)	104 (96.3)	**0.024 ***
200 < PaO_2_/FiO_2_ ≤ 300	35 (8.6)	31 (10.3)	4 (3.7)	1.000
100 < PaO_2_/FiO_2_ ≤ 200	180 (44.0)	148 (49.2)	32 (29.6)	0.466
PaO_2_/FiO_2_ ≤ 100	157 (38.4)	89 (29.6)	68 (63.0)	**<0.001 ***
Mechanical ventilation (days)	14 [7–24]	15 [8–25]	12 [5–19.5]	**0.014 ***
Prone position	263 (62.5)	183 (59.2)	80 (71.4)	**0.022 ***
Continuous muscle blockers	295 (77.4)	210 (75.3)	85 (83.3)	0.095
Tracheotomy	76 (23.3)	64 (27.1)	12 (13.3)	**0.008 ***
Catecholamines	326 (78.2)	230 (75.2)	96 (86.5)	**0.013 ***
ECMO	16 (3.8)	11 (3.5)	5 (4.5)	0.847
Renal replacement therapy	59 (14.0)	31 (10.0)	28 (25.0)	**<0.001***
Thromboembolic events	68 (16.2)	46 (14.8)	22 (19.8)	0.221
**Outcome**				
ICU LOS (days)	17 [8–30]	19 [10–31]	13 [6–24]	**<0.001 ***
In-hospital LOS (days)	26 [13–43]	30.0 [19–48]	13.5 [6–24.3]	**<0.001 ***

Data are all expressed as median [Q1–Q3], mean ± SD or *n*/*N* (%), where *n* is the total number of patients with available data. * *p* < 0.05. Phenotype I is a clinical phenotype corresponding to patients with silent hypoxemia, without signs of ARF. Phenotype II is a clinical phenotype corresponding to patients with both hypoxemia and clinical signs of ARF. Abbreviations: ARDS = acute respiratory distress syndrome, ARF = acute respiratory failure, BMI = body mass index, CT = computed tomography, CRP = C-reactive protein, ECMO = extracorporeal membrane oxygenation, ID = immunodeficiency, ICU = intensive care unit, L = liter, sympt = symptom, LOS = length of stay, O_2_ = oxygen, min = minute, SAPS II = Simplified Acute Physiology Score II.

**Table 2 jcm-11-05034-t002:** Demographics, clinical characteristics, and management according to clinical phenotype.

Demographics	Phenotype I *n* = 186	Phenotype II *n* = 237	*p* Value
Age (years)	65 [57–71.8]	66 [59–72]	0.375
Male	134 (72.0)	177 (74.7)	0.541
**Comorbidities *n* (%)**			
Hypertension	95 (51.1)	140 (59.1)	0.100
BMI > 30 (kg/m^2^)	71 (38.6)	103 (45.0)	0.191
Cardiovascular diseases	49 (26.3)	86 (36.3)	**0.029 ***
Diabetes mellitus	47 (25.3)	71 (30.0)	0.286
Pre-existing renal failure	28 (15.1)	41 (17.6)	0.501
Malignancies or ID	16 (8.6)	36 (15.2)	**0.041 ***
Respiratory diseases	42 (22.6)	53 (22.4)	0.958
**ED management**			
Respiratory rate (/min)	26 [21–30]	30.0 [26–37]	**<0.001 ***
First O_2_ saturation (%)	91.2 [87–95]	89 [82–93]	**<0.001 ***
O_2_ requirement (L/min)	12 [4–15]	15 [9–15]	**<0.001 ***
Heart rate (/min)	93 [83–107]	94.5 [81–106.8]	0.788
Systolic BP (mmHg)	129 [116–140]	130 [114–144]	0.993
Glasgow score scale	15 [15–15]	15 [15–15]	0.448
Temperature (°C)	37.9 [37.1–38.6]	37.8 [37–38.6]	0.612
Shock	7 (3.8)	30 (12.7)	**0.001 ***
Confusion	6 (3.2)	3 (1.3)	0.296
Duration since onset symptom (days)	7 [5–10]	7 [4–10]	0.822
Intubation in the ED	50 (26.9)	109 (46.0)	**<0.001 ***
**Laboratory findings**			
Creatinine (μmol/L)	84 [67–105]	84 [67–105]	0.966
CRP (mg/L)	133.8 [78.7–224.8]	154.4 [92.7–221.5]	0.393
Lymphocytes (/μL)	851.4 ± 434.8	1029.5 ± 840.0	**0.006 ***
pH	7.47 [7.44–7.50]	7.46 [7.41–7.49]	**<0.001 ***
PaO_2_ (mmHg)	67 [56–81]	67 [55–80]	0.648
PaCO_2_ (mmHg)	34 [30–37]	34 [30–39]	0.481
HCO_3_- (mmol/L)	24 [22–26]	23 [20.9–25.4]	**0.013 ***
Lactate (mmol/L)	1.3 [1.0–1.9]	1.5 [1.1–2.1]	0.071
**Radiological findings**			
Typical CT-scan	101 (54.9)	122 (52.4)	0.607
Extension >50%	45 (34.6)	95 (54.3)	**<0.001 ***
**ICU management**			
SAPS II	41 [32–53]	42 [32–55]	0.422
Intubation in the ICU	105 (56.5)	87 (36.9)	**<0.001 ***
ARDS	161 (90.5)	211 (91.3)	0.755
200 < PaO_2_/FiO_2_ ≤ 300	17 (9.6)	18 (7.8)	1.000
100 < PaO_2_/FiO_2_ ≤ 200	72 (40.5)	108 (46.8)	0.583
PaO_2_/FiO_2_ ≤ 100	72 (40.5)	85 (36.8)	1.000
Mechanical ventilation (days)	14 [8–22]	13 [6–25]	0.750
Prone position	118 (63.8)	145 (61.4)	0.622
Continuous muscle blockers	117 (74.5)	178 (79.5)	0.256
Tracheotomy	29 (20.3)	47 (25.7)	0.252
Catecholamines	144 (78.3)	182 (78.1)	0.971
Renal replacement therapy	27 (14.5)	32 (13.5)	0.765
ECMO	5 (2.7)	11 (4.6)	0.296
Thromboembolic events	27 (14.5)	41 (17.5)	0.417
Bacterial coinfection	114 (61.3)	125 (52.7)	0.078
**Outcome**			
ICU LOS (days)	17.5 [9.0–29.8]	17.0 [7.0–30.0]	0.829
In-hospital LOS (days)	26.5 [16.0–42.5]	25.0 [12.0–43.0]	0.267
ICU mortality	35 (18.8)	69 (29.2)	**0.014 ***
In-hospital mortality	35 (18.8)	77 (32.5)	**0.002 ***

Data are all expressed as median [Q1–Q3], mean ± SD or *n*/*N* (%), where *n* is the total number of patients with available data. * *p* < 0.05. Phenotype I is a clinical phenotype corresponding to patients with silent hypoxemia, without signs of ARF. Phenotype II is a clinical phenotype corresponding to patients with both hypoxemia and signs of ARF. Abbreviations: ARDS = acute respiratory distress syndrome, ARF = acute respiratory failure, BMI = body mass index, CT = computed tomography, CRP = C-reactive protein, ECMO = extracorporeal membrane oxygenation, ID = immunodeficiency, ICU = intensive care unit, L = liter, sympt = symptom, LOS = length of stay, O_2_ = oxygen, min= minute, SAPS II = Simplified Acute Physiology Score II.

**Table 3 jcm-11-05034-t003:** Multivariate analysis of the factors associated with clinical phenotype differences.

Demographics	aOR	95%CI	*p* Value
Age (years)	1.001	[0.966–1.036]	0.975
Male	0.619	[0.251–1.528]	0.298
**Comorbidities**			
Hypertension	1.876	[0.776–4.537]	0.162
Cardiovascular diseases	2.302	[0.856–6.189]	0.098
Diabetes	0.400	[0.148–1.083]	0.071
Pre-existing renal failure	0.670	[0.298–1.504]	0.332
Malignancies or ID	0.979	[0.322–2.977]	0.970
Respiratory diseases	0.955	[0.353–2.585]	0.928
**ED management**			
Respiratory rate (/min)	1.102	[1.043–1.166]	**0.001 ***
Glasgow score scale	1.531	[0.918–2.555]	0.103
First O_2_ saturation (%)	0.963	[0.913–1.017]	0.175
O_2_ requirement (L/min)	0.979	[0.900–1.066]	0.632
Heart rate (/min)	1.007	[0.981–1.035]	0.591
Systolic BP (mmHg)	1.006	[0.987–1.024]	0.549
Temperature (°C)	1.521	[1.011–2.288]	**0.044 ***
Shock	4.030	[0.750–21.650]	0.104
Confusion	0.128	[0.008–2.018]	0.144
Duration since onset symptom (days)	0.929	[0.830–1.040]	0.199
Intubation in the ED	3.844	[1.199–12.319]	**0.023 ***
**Laboratory findings**			
Creatinine (>100 μmol/L)	0.325	[0.105–1.008]	0.052
CRP (>100 mg/L)	0.792	[0.323–1.939]	0.609
Lymphocytes (1000/μL)	1.845	[0.799–4.261]	0.151
Lactate (mmol/L)	0.663	[0.384–1.146]	0.141
pH	0.171	[0.000–86.023]	0.578
pO_2_ (mmHg)	1.016	[1.002–1.031]	**0.031 ***
pCO_2_ (mmHg)	1.027	[0.950–1.110]	0.501
HCO_3_- (mmol/L)	0.857	[0.738–0.995]	**0.043 ***
**Radiological findings**			
Typical CT-scan	0.331	[0.128–0.857]	**0.023 ***
Extension > 50%	3.017	[1.270–7.166]	**0.012 ***
**ICU management**			
SAPS II	0.990	[0.961–1.019]	0.492
ARDS			
200 < PaO_2_/FiO_2_ ≤ 300	1.748	[0.354–8.624]	0.493
100 < PaO_2_/FiO_2_ ≤ 200	1.556	[0.457–5.303]	0.479
PaO_2_/FiO_2_ ≤ 100	0.518	[0.123–2.172]	0.368
Mechanical ventilation (days)	1.027	[0.976–1.081]	0.306
Prone position	0.527	[0.211–1.312]	0.169
Renal replacement therapy	1.835	[0.462–7.289]	0.388
ECMO	3.852	[0.516–28.747]	0.188
Thromboembolic events	0.940	[0.321–2.756]	0.911
**Outcome**			
ICU LOS (days)	0.978	[0.928–1.031]	0.405
In-hospital LOS (days)	1.019	[0.981–1.057]	0.330
ICU mortality (days)	2.385	[0.715–7.950]	0.157
In-hospital mortality (days)	3.079	[0.932–10.171]	0.065

* *p* < 0.05. Phenotype I is a clinical phenotype corresponding to patients with silent hypoxemia (no signs of ARF). Phenotype II is a clinical phenotype corresponding to patients with both hypoxemia and clinical signs of ARF. Abbreviations: aOR = adjusted odd ratio, ARDS = acute respiratory distress syndrome, ARF = acute respiratory failure, CT = computed tomography, CRP = C-reactive protein, ECMO = extracorporeal membrane oxygenation, ID = immunodeficiency, ICU = intensive care unit, L = liter, LOS = length of stay, O_2_ = oxygen, min = minute, SAPS II = Simplified Acute Physiology Score II.

## Data Availability

All data generated or analyzed as part of the study are included.

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
