# Peer review of "Silent Hypoxemia in the Emergency Department: A Retrospective Cohort of Two Clinical Phenotypes in Critical COVID-19"

_jcm, 2022, doi:10.3390/jcm11175034_

Round 1
Reviewer 1 Report
Thanks to authors. The authors analyse a very important and actual topic. The additional information suggested by this study will provide original contributions to the literature. One of the strengths of this study is that it was conducted during the COVID-19 pandemic. As stated the authors, this study has several limitations. One of the weaknesses of this article is that it was a retrospective study. Second, the collected data could not be exhaustively detailed owing to crisis circumstances. Third, the size study population is relatively small. Fourth, results of this study might lack statistical power. But, the observations of the clinicians involved in this study clearly bring awareness to the phenotypes that can be seen in COVID-19. This awareness can shed light on clinicians when evaluating patients with COVID-19 in the emergency room, and making the decision to hospitalize these patients in the intensive care unit.
Author Response
Response to reviewers
Thank you for your decision letter related to our manuscript “Silent hypoxemia in the Emergency Department: a retrospective cohort of two clinical phenotypes in critical COVID-19.”
We have much appreciated the constructive comments from the reviewers and we would like to thank you for giving us the opportunity to resubmit it with the review. Please find on the following pages our point-by-point responses to the reviewers’ suggestions. We believe their comments significantly improved the revised manuscript. We will be happy to address any further comments or suggestions you or the reviewers might have.
With best regards,
On Behalf of the authors,
Karine Alamé MD and Pierrick Le Borgne MD
- Reviewer 1
( ) I would not like to sign my review report
(x) I would like to sign my review report
English language and style
( ) Extensive editing of English language and style required
( ) Moderate English changes required
(x) English language and style are fine/minor spell check required
( ) I don't feel qualified to judge about the English language and style
Yes |
Can be improved |
Must be improved |
Not applicable |
|
Does the introduction provide sufficient background and include all relevant references? |
(x) |
( ) |
( ) |
( ) |
Are all the cited references relevant to the research? |
(x) |
( ) |
( ) |
( ) |
Is the research design appropriate? |
(x) |
( ) |
( ) |
( ) |
Are the methods adequately described? |
(x) |
( ) |
( ) |
( ) |
Are the results clearly presented? |
(x) |
( ) |
( ) |
( ) |
Are the conclusions supported by the results? |
(x) |
( ) |
( ) |
( ) |
Comments and Suggestions for Authors
Thanks to authors. The authors analyse a very important and actual topic. The additional information suggested by this study will provide original contributions to the literature. One of the strengths of this study is that it was conducted during the COVID-19 pandemic. As stated the authors, this study has several limitations. One of the weaknesses of this article is that it was a retrospective study. Second, the collected data could not be exhaustively detailed owing to crisis circumstances. Third, the size study population is relatively small. Fourth, results of this study might lack statistical power. But, the observations of the clinicians involved in this study clearly bring awareness to the phenotypes that can be seen in COVID-19. This awareness can shed light on clinicians when evaluating patients with COVID-19 in the emergency room, and making the decision to hospitalize these patients in the intensive care unit.
Dear Reviewer 1,
We are sincerely thankful for the constructive criticisms and valuable comments and hope to shed light on this important topic with our manuscript, it is, indeed, a topic of great interest, and at the
crossroads of several interests.

Reviewer 2 Report
This is a good study on comparison of characteristics, clinical findings, mortality and outcome between silent and symptomatic hypoxemia. Although the manuscript is well written, The study has been conducted Between March 1st and April 30th, 2020 which is long times ago.
Title
1) It is recommended to use no abbreviation in title, secondly the title needs to be clear and concise enough. Phenotype here is not very clear what authors are trying exactly to study.
Result
1) The title "Survivors versus non survivors" can be revised. Its not very clear what is the aim of the title
Discussion
1) The main paragraphs in discussion including limitations and conclusion needs to be significantly revised and improved. Discussion has been written like introduction giving sections and introductory background whereas authors need to summarize their findings, interpret their findings, along with limitations future recommendation can be included. Similarly conclusion should not have introductory information too.
Tables
1) In some titles authors have mentioned the phenotype and somewhere type, It would be helpful if this would be consistent in whole manuscript.
Author Response
Thank you for your decision letter related to our manuscript “Silent hypoxemia in the Emergency Department: a retrospective cohort of two clinical phenotypes in critical COVID-19.”
We have much appreciated the constructive comments from the reviewers and we would like to thank you for giving us the opportunity to resubmit it with the review. Please find on the following pages our point-by-point responses to the reviewers’ suggestions. We believe their comments significantly improved the revised manuscript. We will be happy to address any further comments or suggestions you or the reviewers might have.
With best regards,
On Behalf of the authors,
Karine Alamé MD and Pierrick Le Borgne MD
- Reviewer 2
Open Review
(x) I would not like to sign my review report
( ) I would like to sign my review report
English language and style
( ) Extensive editing of English language and style required
( ) Moderate English changes required
( ) English language and style are fine/minor spell check required
(x) I don't feel qualified to judge about the English language and style
Yes |
Can be improved |
Must be improved |
Not applicable |
|
Does the introduction provide sufficient background and include all relevant references? |
(x) |
( ) |
( ) |
( ) |
Are all the cited references relevant to the research? |
(x) |
( ) |
( ) |
( ) |
Is the research design appropriate? |
(x) |
( ) |
( ) |
( ) |
Are the methods adequately described? |
(x) |
( ) |
( ) |
( ) |
Are the results clearly presented? |
(x) |
( ) |
( ) |
( ) |
Are the conclusions supported by the results? |
( ) |
( ) |
(x) |
( ) |
Comments and Suggestions for Authors
This is a good study on comparison of characteristics, clinical findings, mortality and outcome between silent and symptomatic hypoxemia. Although the manuscript is well written, The study has been conducted Between March 1st and April 30th, 2020 which is long times ago.
Dear Reviewer, 2,
We sincerely thank you for the constructive criticisms and valuable comments, which were of great help in revising the manuscript. We hope our answers, and meet your approval.
Title
- It is recommended to use no abbreviation in title, secondly the title needs to be clear and concise enough. Phenotype here is not very clear what authors are trying exactly to study.
We thank the reviewer for pointing these two inconsistencies out and we have modified the title accordingly: elaborating the abbreviation (emergency department) and clarifying that we are studying the observable clinical characteristics (clinical phenotype) of our cohort.
Result
1) The title "Survivors versus non survivors" can be revised. Its not very clear what is the aim of the title.
Thank you for this remark, the title was revised to ‘survival status’.
Discussion
1) The main paragraphs in discussion including limitations and conclusion needs to be significantly revised and improved. Discussion has been written like introduction giving sections and introductory background whereas authors need to summarize their findings, interpret their findings, along with limitations future recommendation can be included. Similarly conclusion should not have introductory information too.
There are many different ways to write a Discussion section and we have initially opted to discuss the story of silent hypoxemia during the first wave rather than repeat our results. However, we understand the reviewer’s valuable comments and have reconstructed and revised our discussion integrating our summarized findings.
Tables
1) In some titles authors have mentioned the phenotype and somewhere type, It would be helpful if this would be consistent in whole manuscript.
Consistency is indeed key, we thank the reviewer for pointing this out and we corrected these disparities.
